# Study on Bearing Capacity and Failure Mode of Multi-Layer-Encased Geosynthetic-Encased Stone Column under Dynamic and Static Loading

**Bowen Kang [1,2], Jiaquan Wang [2,3,]*, Yuanwu Zhou [2,3,]* and Shibin Huang [2,3]**

1   College of Architecture and Electrical Engineering, Hezhou University, Hezhou 542899, China; kbwen1993@163.com
2   College of Civil and Architectural Engineering, Guangxi University of Science and Technology, Liuzhou 545006, China; hsb321@163.com
3   Guangxi Zhuang Autonomous Region Engineering Research Center of Geotechnical Disaster and Ecological Control, Liuzhou 545006, China
*   Correspondence: wjquan1999@163.com (J.W.); ywzhou@gxust.edu.cn (Y.Z.)

**Abstract:** The "method of overlap" replaces traditional welding to solve the problem of how the geosynthetic-encased stone column is limited by the welding frame during site construction, making the site construction simplified and economical, but its bearing mechanism is not clear. Therefore, the bearing mechanism and failure mode of the stone column was studied through the compression test of the multi-layer geosynthetic-encased stone column under dynamic and static loading. The research shows that the multi-layer encasement improves the modulus and lateral restraint of the stone column, which increases the stress transfer rate and reduces the damage degree of the stone column. The vertical ultimate bearing capacity increase in multi-layer geosynthetic-encased stone columns under dynamic and static loading is significantly different, and the difference can be up to 47.1%; the corresponding number of encasement layers should be selected according to the actual situation. The influence of the difference between dynamic and static loading on the location of the main radial strain of the stone column can be ignored, but the lateral restraint of the stone column under dynamic loading is weakened, the stress transfer rate is reduced, and the radial strain is reduced and more uniform along the stone column height. The vertical ultimate bearing capacity of the one- and two-layer geogrid-encased stone column under dynamic loading is lower than that of static loading. When treating soft foundations, the influence of traffic loads should be considered, and the bearing capacity of the geosynthetic-encased stone column should be appropriately increased in design value.

**Keywords:** geosynthetic-encased stone column; multi-layer encasement; radial strain; failure mode; dynamic and static loading





## 1. Introduction

A GESC (geosynthetic-encased stone column) uses its encasement sleeve to provide lateral restraint for the stone column, offsetting the radial force caused by the bulge of the stone column; it reduces the deformation and maintains the stability of the stone column, and improves the load-bearing performance of OSCs (ordinary stone columns). At the same time, the encasement sleeve reduces the tendency of aggregates to squeeze the soil around the stone column laterally and reduces the mutual influence between the stone column. Because of its excellent engineering properties, such as improving bearing capacity, reducing settlement, and accelerating soft clay consolidation, it has been widely used in the treatment of weak foundations, such as the construction of railways and highways on weak foundations.

Since the concept of GESCs was proposed by Van and Silence [1], some scholars have compared GESCs with OSCs and confirmed that encasement sleeves can improve the bearing capacity, reduce the bulge, and significantly improve the bearing performance of stone columns [2–4]. Later through scaled models, the effects of geogrid strength, the geosynthetic encasement length, the length–diameter ratio, and aggregate size on the bearing performance and deformation of GESCs were studied through a model test or numerical simulation, and the bearing mechanism of GESCs was further studied [5–11]. While studying the bearing performance of single stone columns, some scholars have conducted indoor model test research on GESC composite roadbeds. The test results show that using geosynthetics to encase stone columns can significantly increase the concentrated stress ratio and reduce the settlement of the soft foundation; GESCs have good soft foundation treatment ability [12–14]. Field load tests show that the additional restraint provided by the geogrid improves the stiffness of the stone column, which improves the bearing capacity of the stone column and reduces the settlement of soft soil foundations [15]. To avoid the limitation of test conditions, many scholars studied the shear strain and load transfer mechanism of stone columns by numerical simulation and further explored the reinforcement mechanism of GESCs on soft foundations [16–21]. Few scholars have studied the dynamic load-bearing performance of soft foundation single-stone columns. Yoo C and Abbas Q [22] conducted cyclic loading tests through a scaled model and considered the effects of loading frequency, amplitude, and the geosynthetic encasement length on the bearing performance of the stone column. Their research shows that the overall bearing performance of the stone column in the sand under dynamic loading is greater than that under static loading, and the stress concentration ratio under dynamic loading is smaller than that under static loading. Ardakani et al. [23] conducted a comprehensive parametric study through a 3D finite element simulation. The results showed that increasing the stiffness of encasement to reduce the residual settlement of the geotextile-encased stone column can improve the bearing performance of the stone column under cyclic loading. Zhang et al. [24] carried out cyclic dynamic loading of GESCs through a three-dimensional discrete element model and monitored its deformation characteristics, stress state, and other responses. The experimental results showed that the gradual densification of the crushed stone filler under cyclic loading resulted in a significant increase in the stiffness of the GESC. From the above, it can be seen that the previous studies mainly focus on the bearing performance of GESCs under static load, but when the stone column (OSC or GESC) treats a soft foundation, in addition to the static load of the embankment and roadbed itself, the influence of vehicle load should also be considered [25]. Although the previous test has made some explanations for the bearing performance and mechanism of GESCs under dynamic load, the soil around the stone column is replaced by sand, the boundary conditions of the scaled model are limited, and the simulation error of the numerical simulation causes the test results to have a certain deviation. Additionally, the difference analysis of GESC bearing capacity under dynamic and static loading was not carried out. At the same time, during site construction, the geogrid sleeve needs a large welding frame for section welding, the length is limited by the length of the welding frame, and the transportation cost of the welding equipment is extremely high. Therefore, some scholars proposed the "method of overlap" (that is, the geogrid is rolled into a multi-layer geogrid sleeve according to a certain column diameter, and fixed with nylon ties) instead of welding. Its feasibility has been verified in small- and medium-sized model tests. This method is convenient for construction on site and reduces costs [26], but there are few reports on multi-layer geogrid-encased GESCs under dynamic and static loads.

Given this, this study carried out multiple sets of GESC large-scale indoor dynamic and static loading comparative model tests encased by multi-layer geogrids closer to the actual engineering conditions. The bearing capacity, vertical strain, and lateral earth pressure around the stone column under dynamic and static loading are monitored in real time, and the final radial strain is measured after the test. The influence of the number of cycles, loading method, number of encasement layers, etc., on the bearing performance

of GESCs was analyzed, and the bearing mechanism of multi-layer-encased geosynthetic-encased stone columns was further understood by comparing the bearing difference of stone columns under dynamic and static loading, which provided the experimental basis for GESC engineering design.

## 2. Model Test Design

### 2.1. General Situation of Test

In practical engineering, the aggregate size of GESC is 50~200 mm [27], the stone column diameter varies from 600 mm to 1000 mm, and the diameter–length ratio of the stone column varies between 5 and 20 [28]. By referring to the similarity ratio selection method in Baker WE [29] and Yoo C and Abbas Q [22], and comprehensively considering the properties, quality, and other factors of GESC in the test, the scale effect of the model is minimized. The similarity ratio $\lambda$ is 4 ($\lambda$ = model: prototype), and the scale factor of geogrid strength is $\lambda^2 = 16$. For a stone column diameter of 200 mm, corresponding to the site size of 800 mm, the aggregate size is between 20 and 50 mm which was considered representative of typical conventional stone column aggregate [26]. The geogrid in this study is a biaxial geogrid commonly used in construction sites, and its strength is usually from 100 to 400 kN/m. Considering the size effect, the strength of geogrid is still in the typical range.

Six groups of experiments were designed, as shown in Table 1. $S_2$, $S_3$, $D_2$, and $D_3$ in the table are multi-layer encasements; that is, biaxial geogrid around the equal diameter PVC pipe 2, 3 layers, respectively, and overlap the width of three geogrid holes, and finally tied with a rolling strip.

**Table 1.** Compression test scheme.

| Test No. | No. of Encased Layers | Column Length (mm) | Column Diameter (mm) | Particle Size (mm) | Loading Method |
|----------|----------------------|--------------------|----------------------|--------------------|----------------|
| $S_1$ | 1 | | | | |
| $S_2$ | 2 | 800 | 200 | 20~50 | static loading |
| $S_3$ | 3 | | | | |
| $D_1$ | 1 | | | | |
| $D_2$ | 2 | 800 | 200 | 20~50 | dynamic loading |
| $D_3$ | 3 | | | | |

### 2.2. The Test Equipment and Instruments

The indoor scale model test was loaded by the DJM-500 multi-functional servo-controlled hydraulic loading system of Guangxi University of Science and Technology, and the test results were collected through the data acquisition system. To simulate the real boundary conditions as much as possible and reduce the size effect, it was carried out in a 1500 mm × 1600 mm × 2000 mm (length × width × height) model box, as shown in Figure 1. Figure 2 is the layout of the instrument. The top settlement of the stone column, vertical load, and lateral earth pressure around the stone column was monitored through the pressure sensor, displacement sensor, and earth pressure box. Due to the fragmentation of aggregate, stress concentration occurred easily between the aggregate. To accurately obtain the pressure at the top and bottom of the stone column, and understand the variation law of the stress transfer rate of the stone column, three earth pressure boxes were arranged at the bottom and top of the column to obtain the average value of the earth pressure, and an equal thickness sand cushion was placed on the upper and lower sides of the earth pressure box to prevent it from being punctured by aggregate during the vertical compression, and at the same time, make the stress evenly transmitted on the earth pressure box.

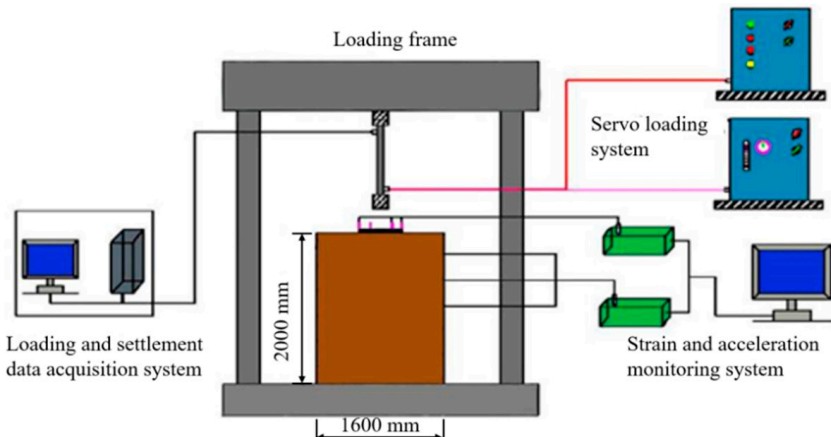

**Figure 1.** DJM-500 servo-controlled hydraulic loading system.

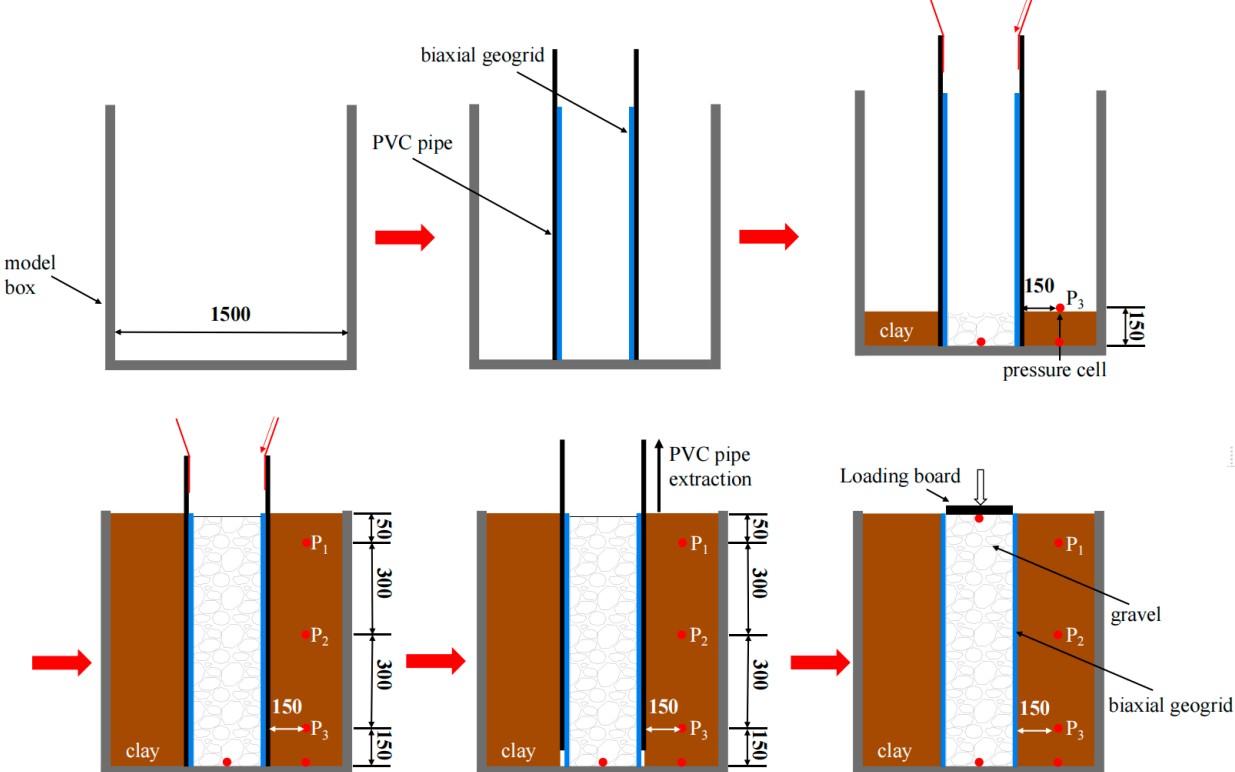

**Figure 2.** Flow chart of dynamic load test of geosynthetic-encased stone columns.

### 2.3. Test Material

The soft foundation used in this study was prepared from local red clay in Liuzhou, and its basic mechanical parameters are shown in Table 2. The aggregate was selected from the naturally graded aggregate in the quarry in Liuzhou, Guangxi, and its gradation is shown in Figure 3. The encasement sleeve was made of polypropylene bidirectional plastic geogrid, which was made by cutting, bending and shaping, and overlapping sections. The diameter and height of the geogrid sleeve were 200 mm and 800 mm, respectively. The geogrid section was overlapped with 3 mesh, and self-locking nylon cable ties were used in three rows to tie. The total tensile resistance of the tie band at the joint was designed to be equal to the total tensile resistance of the geogrid section. It was verified by the unconfined pre-test of a single stone column that the lapping method met the test requirements. All geogrid-specific technical indicators are shown in Table 3.

**Table 2.** Basic mechanical parameters of soft soil.

| Parameters | Value |
|---|---|
| Liquid limit, $w_L$(%) | 62.40 |
| Plastic limit, $w_p$(%) | 32.20 |
| Plasticity index, $I_p$(%) | 30.20 |
| Specific gravity, (-) | 2.70 |
| Water content, $w$(%) | 54.20 |

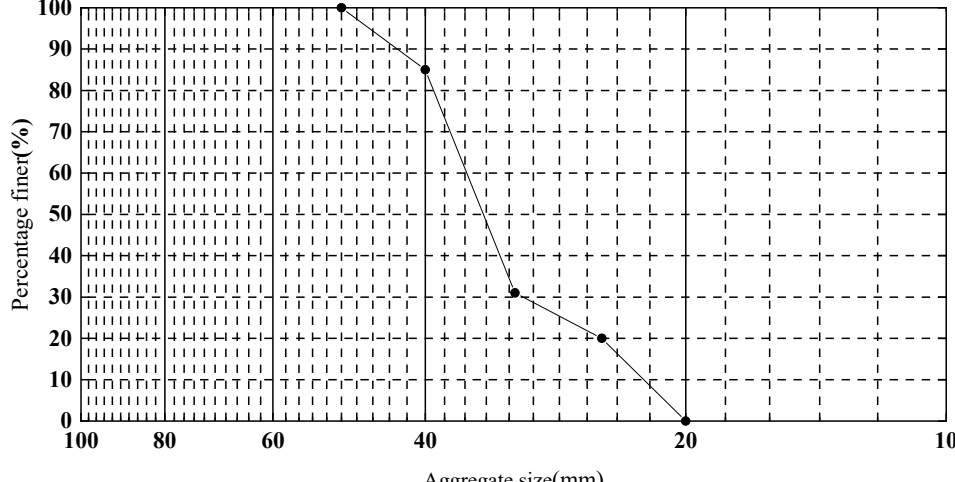

**Figure 3.** Grading curves of aggregate.

**Table 3.** Specific technical indicators of geogrid.

| Item | Value |
|---|---|
| Longitudinal tensile yield force per meter (kN/m) | 18.6 |
| Longitudinal rib width (mm) | 3.0 |
| Longitudinal rib thickness (mm) | 1.5 |
| Transverse tensile yield force per meter (kN/m) | 15.4 |
| Transverse rib width (mm) | 2.0 |
| Transverse rib thickness (mm) | 1.0 |
| Node size (mm) | 4 × 3 |
| Mesh size (mm) | 20 × 20 |

### 2.4. Test Filling and Loading Mode

To prevent water loss during the test and keep the moisture content constant, a transparent plastic film was laid on the inner wall of the model box, and the overlapping part of the plastic film was adhered with tape to make the plastic film close to the inner wall of the model box (as shown in Figure 4). At the same time, the color strips were laid on the plastic film in the same way, to prevent the film from being damaged and increase the anti-seepage performance of the model box. After the plastic film was laid, the geosynthetic encasement was placed in a fixed position by a laser level (as shown in Figure 5). To ensure the compactness and uniformity of the column, the quartering method was adopted for the sample, and the aggregate with a total mass of 46 kg was loaded into the geosynthetic encasement in 4 layers. Each layer of aggregate was compacted by free fall motion with a vibrating rod at the height of 200 mm, and was compacted 25 times so that the density of aggregate was controlled at 1.83 g·cm$^{-3}$. While vibrating and mashing the aggregate, the position of the geosynthetic encasement was kept. The above steps were repeated until the filling height reached 800 mm.

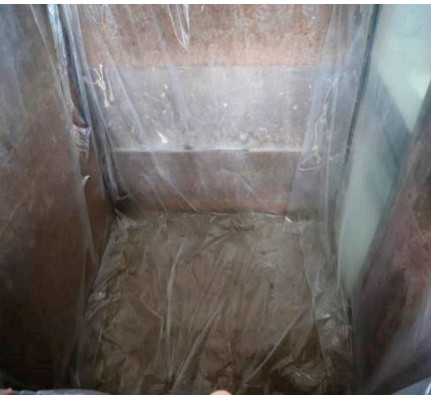

**Figure 4.** Waterproof laying of model box.

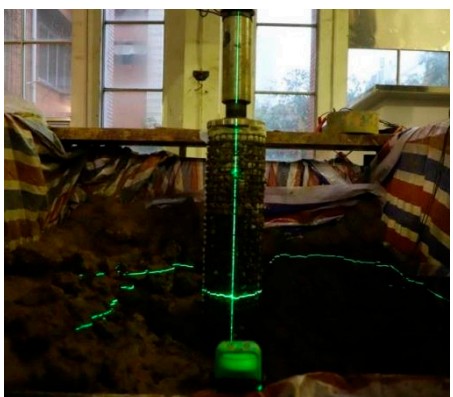

**Figure 5.** Schematic diagram of GESC position fixation.

Then, the soft soil was filled layer by layer; each layer's thickness was 200 mm. Before the soft soil filling was completed, the soft soil was sampled 5 times, and the error between the measured and expected moisture content was controlled within 2%. When filling the soft soil, because the soft clay had fluid plasticity, it was inconvenient to use tools such as weights for compaction. Therefore, a color band cloth was laid on the surface of soft clay, and the uniformity of each layer was ensured by a cone penetrometer after manual rubbing and pressing. After filling to the specified height, the interface was leveled and the instrument was placed, and the above steps were repeated to continue filling until the soft soil layer was flush with the top of the GESC. After the completion of the test model, to avoid water evaporation and maintain the uniform water content of the soft soil, the plastic film was covered on the surface of the soft soil and stood for 12 h to make the soft soil fully consolidated and settled.

This test adopts the step-by-step equal loading method, and the applied load for each stage was 1/10 of the pre-test ultimate bearing capacity, of which the first-stage loading was twice the graded load. The ultimate bearing capacity of the stone column was 20 kN, measured by pre-test. During the loading process, the first stage load was 4 kN, and each stage load was 2 kN thereafter. After reaching the predetermined load of each stage, a dynamic load was applied at a constant frequency, and the dynamic load amplitude was 0.5 kN; each stage load was loaded for 15 min until the GESC was damaged. The application of multi-stage load in this study was different from that in the study of Aqoub [30] and Tang [31]. To avoid the instability of the stone column under the impact of sudden increased load when entering the next stage load, this test adopted a uniform loading method for slow loading to ensure the authenticity of the test results. Figure 6 is the dynamic and static load diagram.

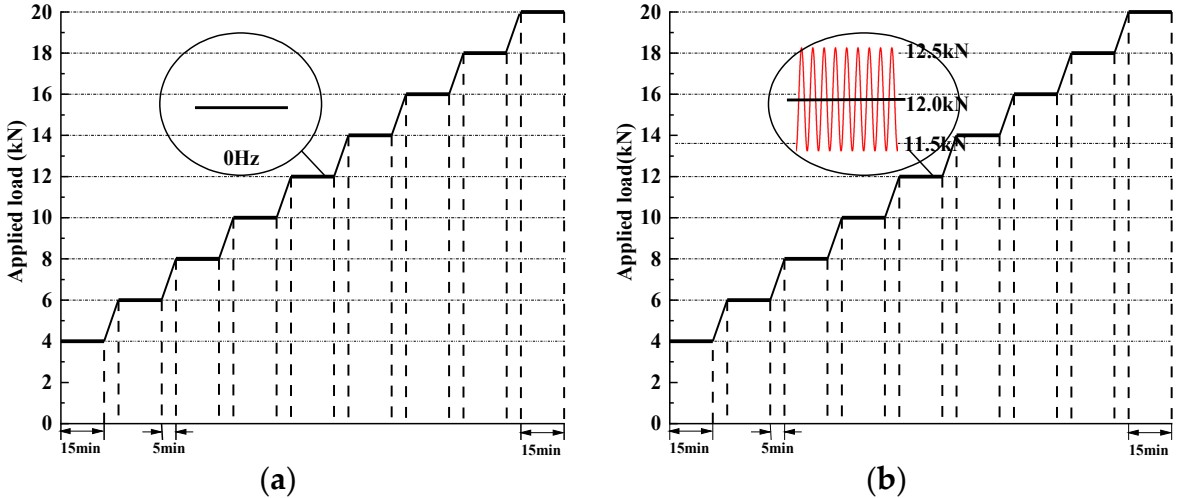

**Figure 6.** Loading scheme adopted. (**a**) static loading. (**b**) dynamic loading.

## 3. Experimental Results

### 3.1. Compression Test of Single GESC under Dynamic Loading

#### 3.1.1. Influence of Cycle Number on Bearing Capacity

Figure 7 shows the variation curve of settlement of GESC tops with $N$ (the number of cycles). The stone column was loaded with dynamic loading for 15 min at each stage and loaded until the GESC was destabilized and damaged. As can be seen from Figure 7, when $N \leq 5880$, the $s$–$N$ curves of $D_1$ and $D_2$ are basically the same, but with the increase in $N$, the difference in $s$–$N$ curves of $D_1$ and $D_2$ is gradually significant, which indicates that when the number of encased layers is small, the influence of a certain degree of cycle number on the stone column is negligible. The settlement of the GESC increases and tends to be linearly compressed as $N$ increases, which was because the aggregate gradually formed a dense and stable skeleton under the action of dynamic loading, and the stone column developed from dense compression to elastic compression. As $N$ continued to increase, the settlement continued to increase until the GESC was unstable and damaged. The final settlement of $D_1$, $D_2$, and $D_3$ was 94.95 mm, 123.43 mm, and 156.26 mm, respectively, which increased by 21.01% and 39.24% for $D_2$ and $D_3$, respectively, compared to $D_1$. The higher settlement of $D_2$ and $D_3$ compared to $D_1$ is attributed to the restraint of multi-layer encasement. The multi-layer encasement not only indirectly improved the overall stiffness of the encasement sleeve, which made the geogrid less likely to yield to damage, but the overlapping part of the external geogrid could also fix the protruding aggregate due to the damage of the internal geogrid, which improves the overall stability and delays the failure of the GESC.

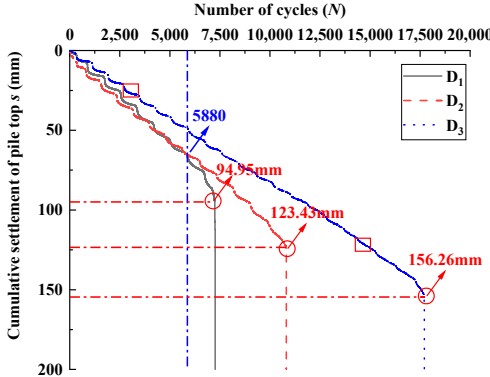

**Figure 7.** Dynamic loading $s$–$N$ curve of a single GESC.

### 3.1.2. Influence of Encasement Layer on Bearing Capacity

To study the influence of the number of encasement layers on bearing capacity, the lateral limit dynamic loading tests were conducted on the GESC with different numbers of encasement layers. Figure 7 shows the vertical load-vertical strain relationship curves of the GESC with different numbers of encasement layers under dynamic loading. From Figure 8, it can be seen that ① under the same vertical dynamic loading, the vertical strain of $D_2$ and $D_3$ is smaller than the vertical strain of $D_1$, and the difference in vertical strain increases significantly with the increase in vertical load, which indicates that the number of encasement layers can significantly improve the bearing capacity of GESCs. ② Under dynamic loading, $D_1 \rightarrow D_2 \rightarrow D_3$, the ultimate bearing capacity of the stone column is 652.6 kPa $\rightarrow$ 908.4 kPa $\rightarrow$ 1432.1 kPa, and the ultimate vertical strain is 11.9% $\rightarrow$ 14.8% $\rightarrow$ 19.8%. The ultimate bearing capacity of $D_2$ and $D_3$ are increased by 38.7% and 119.4%, and the ultimate strain is increased by 24.4% and 66.4%, respectively, compared with $D_1$. The ultimate bearing capacity and ultimate vertical strain of GESCs increase with the increase in encasement layers, which is because after the stone column is compressed to a certain degree, the stone column starts to bulge, the inner and outer geogrid gaps are squeezed dense by the aggregate, and the outer geogrid starts to provide additional lateral binding force for the stone column, which indirectly improves bearing capacity for the stone column. When the vertical load continues to be applied, part of the inner geogrid reaches the ultimate tensile strength and starts to be destroyed, but the outer geogrid of the encasement sleeve still provides restraint and supports the stone column to withstand the vertical load until the outer geogrid starts to reach the ultimate tensile strength and is destroyed. Therefore, the vertical strain at failure of multi-layer GESCs is higher than that of single-layer GESC failures.

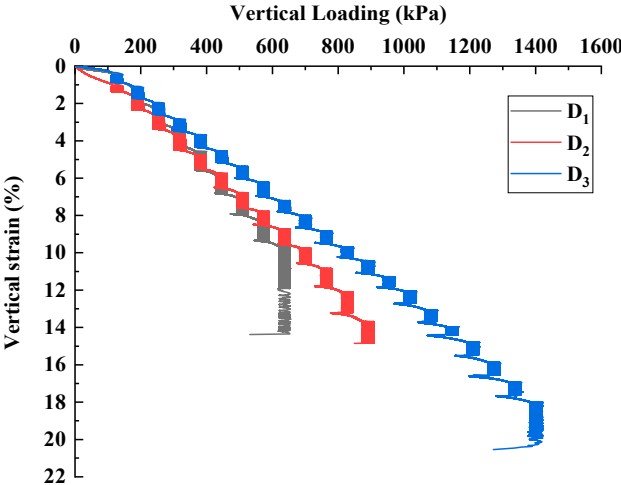

**Figure 8.** Vertical loading–vertical strain curve.

Meanwhile, to analyze the reasons for the significant differences in vertical strains of stone columns with different numbers of encasement layers and to further understand the reinforcement mechanism of multi-layer-encased GESC under dynamic loading, the percentages of settlement and total vertical strains at different load stages of the stone column (i.e., the ratio of settlement in pre-cyclic and cyclic stages, where the percentage of pre-cyclic and cyclic section settlement is the percentage of total pre-cyclic and cyclic settlement relative to the total settlement under dynamic loading at all levels) and the settlement under dynamic loading at a specific single stage are quantified and analyzed as shown in Figures 9 and 10. It can be seen that from Figure 9, when $D_1 \rightarrow D_2 \rightarrow D_3$, the percentage of the settlement in the pre-cycle stage is 23% $\rightarrow$ 28% $\rightarrow$ 28%, and its settlement is 21.59 mm $\rightarrow$ 33.78 mm $\rightarrow$ 43.59 mm. The percentage of the settlement in the cycle stage is 77% $\rightarrow$ 72% $\rightarrow$ 72%, and its settlement is 72.89 mm $\rightarrow$ 85.01 mm $\rightarrow$ 110.72 mm. Relative to $D_1$, the settlement and percent settlement of $D_3$ in its pre-cyclic stage increased by 21.7%

and 101.9%, respectively, while the percent settlement in the cyclic stage decreased by 6.5% and the settlement increased by 51.9%. As can be seen from Figure 12, with the increase in encasement layers, the vertical strain difference under each vertical load of the stone column before cycling is not much different, ranging from 0.20% to 0.29%, while the vertical strain difference under each vertical load during the cycle stage is not much different. The strain difference decreased significantly, from 1.05% to 0.70%, resulting in a decrease in the proportion of stone column settlement in the entire compression process during the cycle stage.

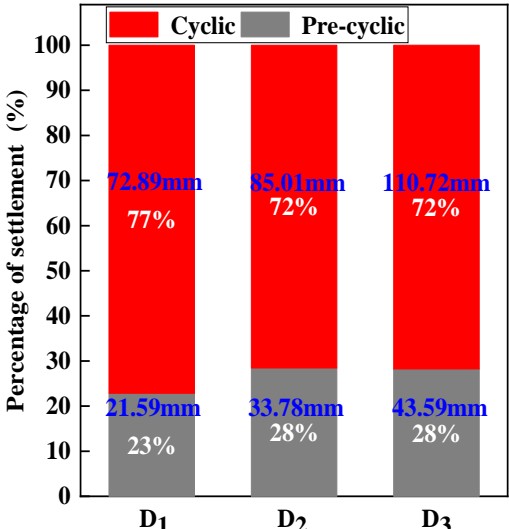

**Figure 9.** The settlement with different encasement.

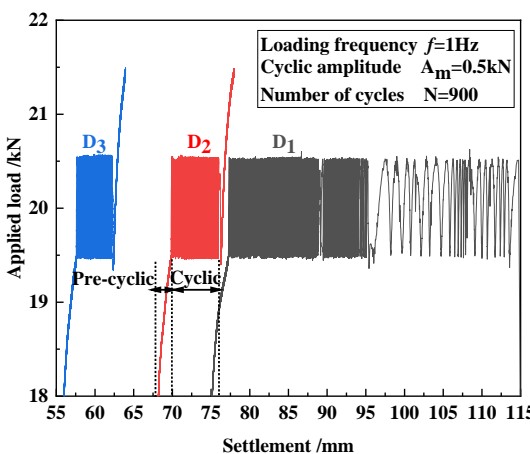

**Figure 10.** Settlement under dynamic loading (20 kN) layer under dynamic loading.

### 3.2. Compression Test of Single GESC under Static Loading

Figure 11 shows the vertical load–strain curves of the OSC and GESC under static loading. From Figure 11, it can be seen that the ultimate vertical strains of $S_0$ and $S_1$ are 6.4% and 10.2%, respectively, and the vertical ultimate bearing capacity is 127.3 kPa and 686.6 kPa, respectively, when the GESC is destabilized. The ultimate vertical strain and vertical ultimate bearing capacity of $S_1$ are 1.6 times and 5.4 times of $S_0$, respectively, which shows that the lateral restraint provided by GESC can significantly reduce its vertical strain and improve the vertical bearing capacity [7]. Meanwhile, the ultimate vertical strains of $S_1$, $S_2$, and $S_3$ were 10.2%, 14.9%, and 20.5%, respectively, and the ultimate vertical loads were 686.6 kPa, 1092.1.0 kPa, and 1423.4 kPa, respectively; the ultimate vertical strains of $S_2$ and $S_3$ were increased by 46.1% and 101.0%, respectively, and the ultimate vertical loads were increased by 59.1% and 107.3%. It can be seen that the additional lateral restraint provided

by the multi-layer encasement can still result in a significant increase in the load-bearing performance of the GESC relative to the single-layer encasement.

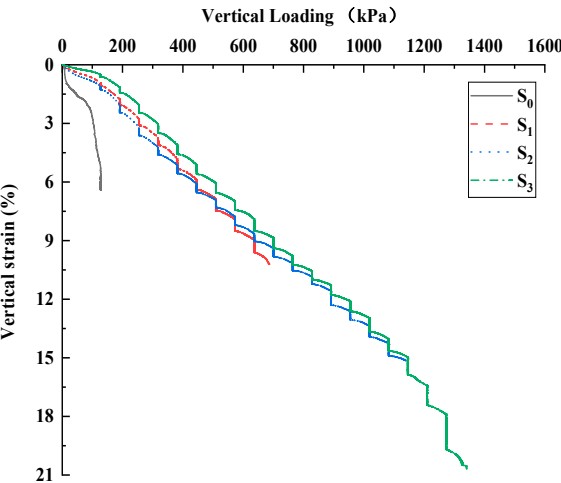

**Figure 11.** Variation of vertical load with strain for static loading.

To explore the reasons for the significant difference in the bearing performance of GESCs with different encasement layers under static loading, and to further understand the reinforcement mechanism of multi-layer geogrid-encased GESCs under static loading, the settlement percentages of GESCs with different encasement layers at different loading stages (i.e., the ratio of settlement in pre-cyclic and cyclic stages, where the percentage of pre-cyclic and cyclic section settlement is the percentage of total pre-cyclic and cyclic settlement relative to total loading settlement under dynamic loading at all levels) were analyzed. It can be seen from Figure 12 that when $S_1 \to S_2 \to S_3$, the settlement percentage of its pre-load stage is 57%→53%→43%, and its settlement is 44.15 mm→63.19 mm→68.34 mm. During the load stage, the settlement percentage was 43%→47%→57%, and the settlement was 32.69 mm→55.99 mm→89.36 mm. Compared with $S_1$, the settlement of $S_3$ in the pre-load stage increased by 54.8%, and the settlement percentage decreased by 24.6%. In the loading stage, the settlement percentage and settlement increased by 32.6%, and the settlement amount increased by 173.4%, respectively.

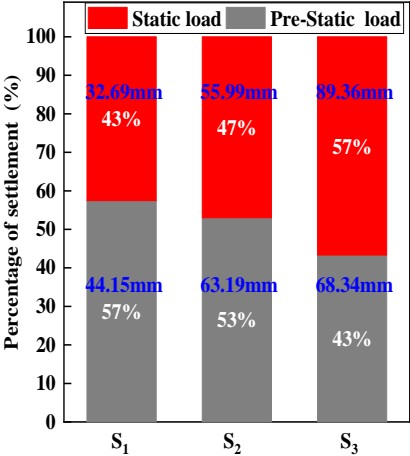

**Figure 12.** The effect of encasement layers on settlement percentage.

*3.3. Comparative Analysis of Load-Bearing Performance under Dynamic and Static Loading*

3.3.1. Analysis of Stone Column Settlement

To further explore the bearing mechanism of GESC under dynamic and static loading, and to explain the reasons for the significant differences in the percentage of stone column

settlement between the pre-load stages and during load stages in Figures 8 and 11, the stone column top settlement data of pre-load stages and during load stages under all levels of loading were extracted from Figures 8 and 10, and the vertical load–vertical strain difference graphs (i.e., the vertical strain differences corresponding to the pre-load stages and during load stages for all levels of loading) were plotted for data analysis. The data are shown in Figure 13. From Figure 13a,b, it can be seen that in the pre-load stages, the vertical strain difference value under the first loading load (128 kPa) is much larger than that under the second loading load, which is because the vertical load is applied in a step-by-step equal loading mode, and the first loading load is twice as large as the graded load [32]. Meanwhile, as observed in Figure 13a, the vertical strain difference value gradually decreases with the increase in vertical load step by step in the pre-static loading stage, which is the result of the gradual compacting of the aggregate under the vertical load. During the loading process, the vertical strain difference value first increases and then maintains in the range of 0.27~0.48%. After the vertical load reaches the ultimate bearing capacity, the bearing capacity of the stone column decreases, and its vertical strain difference value increases under the same vertical load increment. As observed in Figure 13b, the vertical strain difference value tends to be the same in the pre-load stage of dynamic load and maintains within the range of 0.20~0.29%. During the loading cycle, the vertical strain difference values of $D_1$ and $D_2$ were maintained within a certain range, and when the applied load reached the ultimate bearing capacity of the stone column, the vertical strain difference value increased, while the vertical strain difference value of $D_3$ was basically maintained at 0.62%, which was linear compression, indicating that the stone column stiffness increased by increasing the number of encasement layers. In Figure 13, the difference in the vertical strain of the stone column under dynamic and static loading is significantly different. The reason is that in the pre-loading stage, although the increment of vertical load applied at each stage is equal, the vibration effect of the dynamic loading is transmitted to the interior of the stone column in the form of a power wave compared to static loading, which reduces the friction between the aggregates and makes it easier to move between the aggregates under the vertical force and fill the gap to achieve a dense state. Therefore, the GESC is denser than that under static loading after the dynamic loading, which makes the column under the monotonic load in the next stage before the loading phase, the vertical strain of the column is smaller relative to the static loading, while in the loading phase, the vertical strain of the column is larger relative to the static loading. Meanwhile, in the loading stage, the dynamic loading continuously impacts the aggregates and the soil around the GESC, and its dynamic loading impact makes the aggregates dense and the soil around the GESC loose, which also intensifies the vertical compression of the column, making the dynamic loading vertical strain difference value larger than the static loading.

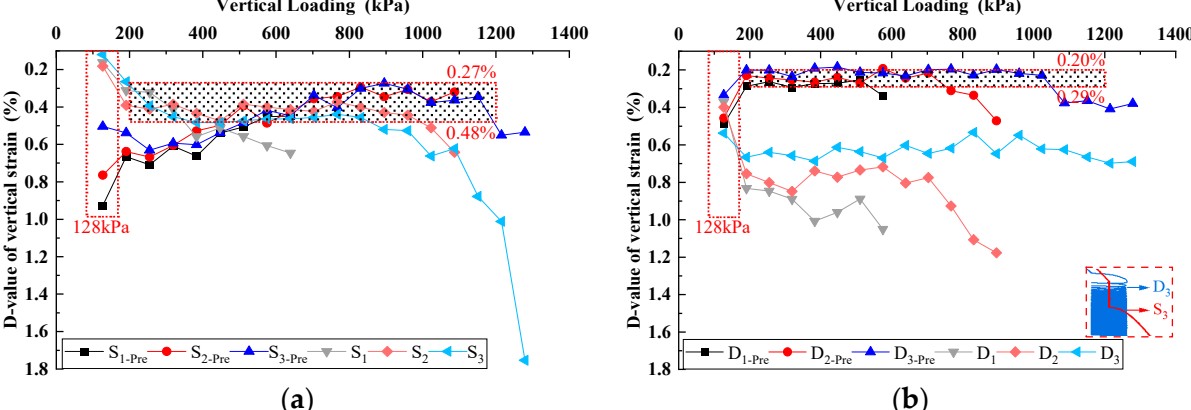

**Figure 13.** Vertical load–vertical strain difference curve. (**a**) Static loading. (**b**) Dynamic loading.

### 3.3.2. Analysis of Stone Column Failure Mode

To explore the influence of the difference between dynamic loading and static loading on the failure mode of the GESC, after the indoor test, the GESC placed in soft soil was half-excavated, and the soft soil section was cleaned and leveled by a geotechnical shovel. After that, the failure mode of the GESC was photographed by a Canon EOS 6D Mark II HD SLR camera. After filming, the GESC was fully excavated, and the radial strain of the stone column and its position height were measured and recorded. The difference in ultimate bearing capacity and radial strain of the GESC under dynamic and static loading was compared to evaluate the effect of dynamic loading on the radial deformation of the GESC. Figures 14 and 15 show the failure mode and bulging strain at different depths under dynamic loading, respectively. Among them, Figure 15 is the simplified bulging strain diagram, which only shows the typical bulging strain of the stone column shown in Figure 14.

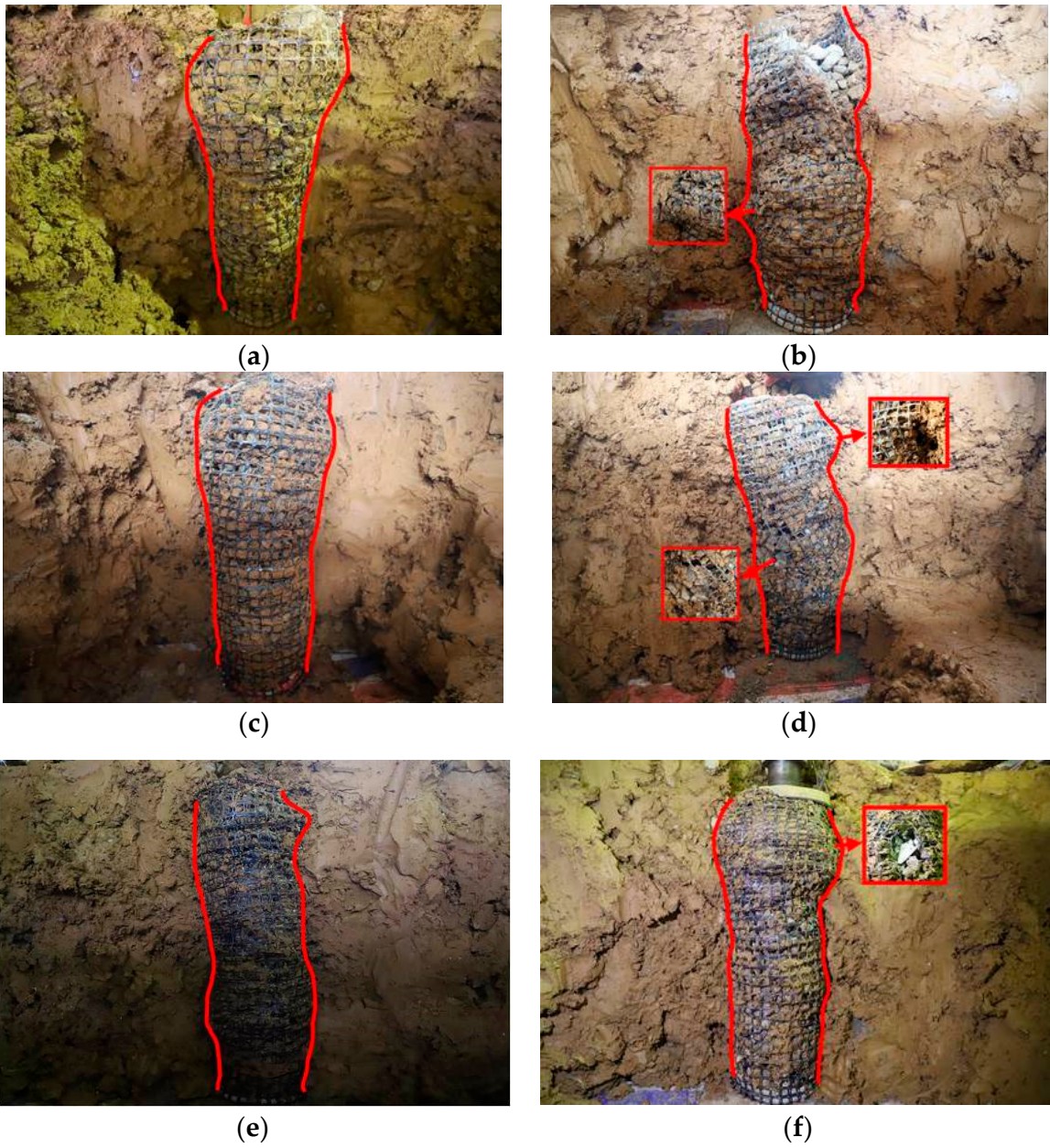

**Figure 14.** Failure mode of geosynthetic-encased stone column under dynamic loading. (**a**) $S_1$. (**b**) $D_1$. (**c**) $S_2$. (**d**) $D_2$. (**e**) $S_3$. (**f**) $D_3$.

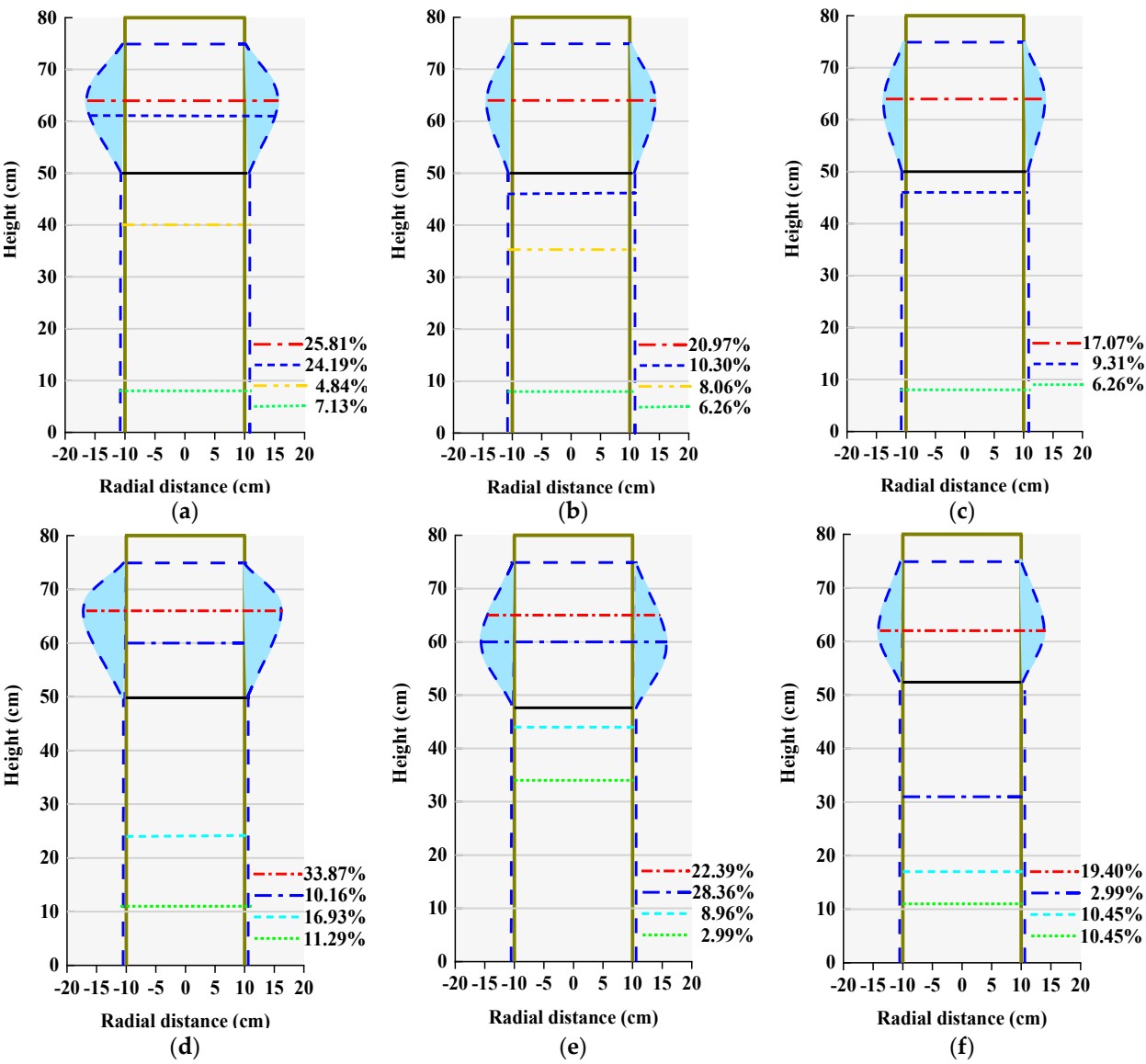

**Figure 15.** Bulging position of GESC. (**a**) $D_1$. (**b**) $D_2$. (**c**) $D_3$. (**d**) $S_1$. (**e**) $S_2$. (**f**) $S_3$.

From Figure 14, it can be seen that ① the measured radial strains are consistent with the results of Hong et al. and Hugher and Withers [33,34]: the GESC radial strain mainly occurs in the upper half of the stone column, and the maximum radial strain occurs at a depth of two to three times the column diameter from the top of stone column, while there is a slight uniform lateral bulge at the bottom of the column. Under dynamic and static loading, the GESC failure mode is bulging damage, and the stone column goes from the compression-density stage–linear elastic stage–yielding stage–failure stage. When the vertical load continues to increase, the gap between the aggregate is continuously compacted, and the stone column generates radial strain. When the geogrid reaches the ultimate tensile strength, the stone column generates plastic deformation, and the top geogrid splits and tears, which eventually leads to stone column destabilization and damage. ② According to the measurement, when the stone column is damaged, the longitudinal splitting failure of the geogrids of $S_1$ and $S_2$ is 21 cm and 22 cm, and the longitudinal tearing failure of the geogrids of $S_3$ is 15 cm. The failure length of $S_3$ is 28.6% lower than that of $S_1$, and the longitudinal splitting failure length of $D_1$, $D_2$, and $D_3$ is 26 cm, 16 cm, and 14 cm. The splitting length of $D_3$ is 42.6% lower than that of $D_1$. It can be seen that the longitudinal splitting failure length of geogrid under dynamic loading is longer than that under static

loading. At the same time, as the number of encasement layers increases, the difference between dynamic and static loading on the failure mode of the stone column decreases.

From Figure 15, we can see that①under dynamic and static loading, the radial strain of the GESC decreases with the increase in depth from the top of the stone column, but the maximum radial strains in the upper and lower parts of $S_1$, $S_2$, and $S_3$ is 22.58%, 25.37%, and 16.41%, respectively, while the difference of maximum radial strain in the upper and lower parts of $D_1$, $D_2$, and $D_3$ is 20.97%, 14.71%, and 10.81%, respectively. It can be seen that the radial strain caused by the dynamic loading is more uniform along the stone column height compared to static loading.②The maximum radial strains of $D_2$ and $D_3$ are 20.97% and 17.07%, respectively, which are 18.87% and 26.89% lower than that of $D_1$, and the overall radial strain of $D_2$ and $D_3$ decreases. At the same time, according to Figures 14 and 15, it can be seen that the failure modes of the GESC under dynamic loading and static loading are different to some extent, but the failure modes are all bulging failure, and the main radial strain positions are similar. The influence on the strain position can be ignored.

### 3.3.3. Analysis of Lateral Earth Pressure around Stone Column

The lateral restraint of soil around the stone column has a significant effect on the deformation of the GESC, and the lateral soil pressure can provide a better understanding of the bearing mechanism of the stone column under dynamic and static loading. Due to the similar development law of soil pressure around GESCs with different encasement layers under dynamic and static loading, due to space limitations, we selected the $S_3$, $D_1$, and $D_3$ test groups with representative measurement points $P_1$, $P_2$, and $P_3$ for lateral soil pressure development law analysis, as shown in Figure 16. It can be seen from Figure 16a that under the static and dynamic loading, the lateral earth pressure $P_2 > P_1 > P_3$, which is caused by the extrusion of the column perimeter soil by the uneven radial strain occurring along the column at different heights; $P_2$ is near the location where the maximum radial strain occurs in GESC, so its value is the largest. Meanwhile, at the same location and under the same vertical load, the lateral soil pressure induced by the static loading is larger compared to the dynamic loading; e.g., at $P_2$, under 20 kN vertical load, $S_3 \rightarrow D_3$, the lateral soil pressure changed from 1.91 kPa to 0.99 kPa, a decrease of 48.2%, which is the result of the dynamic loading weakening the column perimeter soil, consistent with Zhang et al. [35]. From Figure 16b, it can be seen that at the same location and under the same vertical load, the lateral soil pressure decreases as the number of encasement layers increases; for example, at $P_2$, under 20 kN vertical load, $D_1 \rightarrow D_3$, the lateral soil pressure changes from 7.47 kPa to 0.99 kPa, a decrease of 86.7%, which indicates that the encasement sleeve plays a significant restraining role.

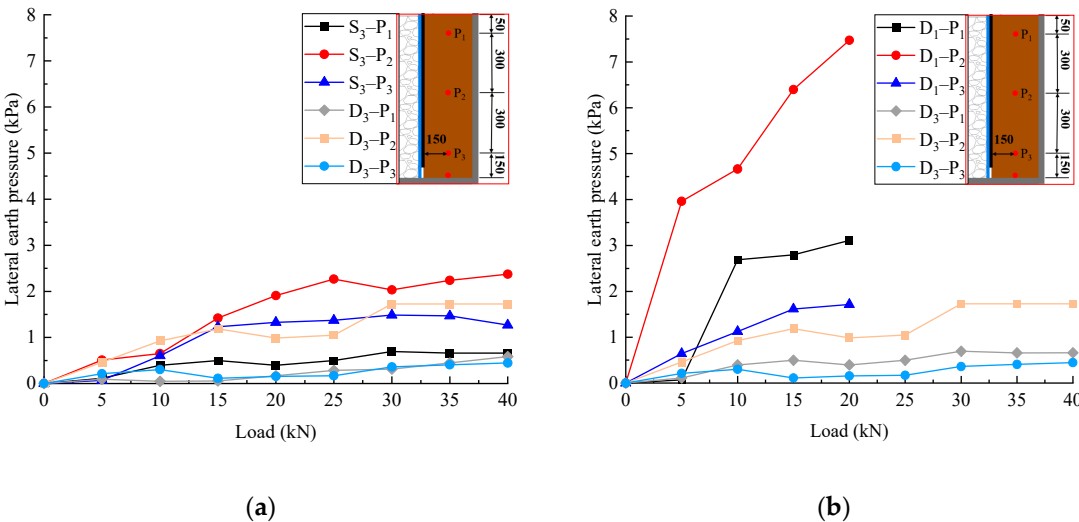

**Figure 16.** Lateral pressure beside GESC. (**a**) $S_3$ and $D_3$. (**b**) $D_1$ and $D_3$.

### 3.4. Analysis of GESC Bearing Mechanism under Dynamic and Static Loading

When GESC is used for soft foundation treatment, its bearing performance is mainly affected by the geogrid stiffness, column length, column diameter, and aggregate size [35], and the core of its influence is the stone column modulus under lateral restraint. The modulus of GESC under lateral restraint is related to the geogrid strength and the relative density of the aggregate [36], and when the geogrid strength and the relative density of the aggregate are the same, the column modulus depends on the lateral frictional resistance of the column perimeter and the modulus of the stone column itself. From the real-time variation law of the pile stress transfer rate of the pile body under dynamic and static loading in Figure 17, it can be seen that the stress transfer rate of GESC under dynamic loading is less than that of static loading, which is because dynamic loading, relative to static loading, has its effect transferred to the interior of the stone column in the form of a dynamic wave and spreads to the column perimeter. Under the vibration, the soil around the column is disturbed and weakened, as shown by Zhang et al. [35], and the lateral restraint effect on the column is weakened, which indirectly leads to the modulus of the GESC decreasing and the stress transfer rate decreasing, and the experimental results are shown in Table 4. At the same time, the difference in lateral restraint effect on the column along the column height decreases, so that the filler expands more uniformly along the column height in all directions, and the overall is more uniform and less bulging. In contrast, under static loading, the upper part of the column is subject to less restraint than the lower part under the confinement of the dense column perimeter soil, so the upper part of the column is more likely to bulge and bulge to a greater extent under static loading. The macroscopic phenomenon of column bulging in Figures 14 and 15 confirms this view.

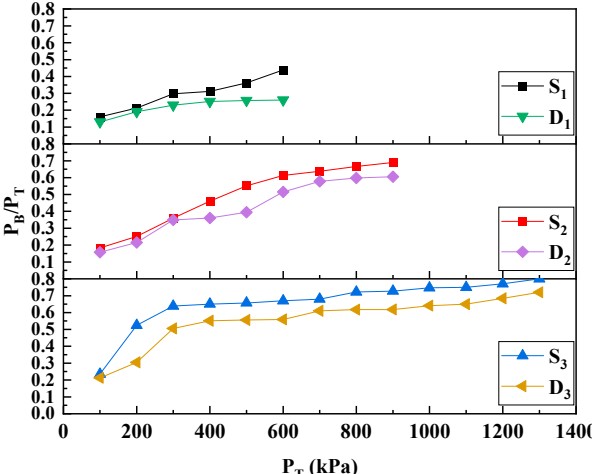

**Figure 17.** Real-time variation curve of GESC stress transfer rate under dynamic and static loading.

**Table 4.** Summary of test results.

| Test No. | Loading Frequency (Hz) | Maximum Radial Strain (%) | Ultimate Load (kPa) | Ultimate Vertical Strain (%) | Modulusl (MPa) |
|---|---|---|---|---|---|
| $S_1$ | 0 | 33.87 | 686.6 | 10.2 | 67 |
| $S_2$ | 0 | 28.36 | 1140.0 | 15.2 | 75 |
| $S_3$ | 0 | 19.40 | 1247.4 | 17.6 | 71 |
| $D_1$ | 1 | 25.81 | 652.9 | 14.3 | 46 |
| $D_2$ | 1 | 20.97 | 893.5 | 16.2 | 55 |
| $D_3$ | 1 | 17.07 | 1421.9 | 19.9 | 71 |

Note: The modulus of GESC is defined as $E_c = p_c / \varepsilon_v$; $p_c$ is the vertical pressure, and $\varepsilon_v$ is the vertical strain. The vertical strain is obtained from Figure 12.

As can be seen from Figure 18, when the vertical stress is 400 kPa, the stress transfer rates of $S_1$, $S_2$, and $S_3$ are 0.31, 0.46, and 0.65, respectively, while the stress transfer rates of $D_1$, $D_2$, and $D_3$ are 0.25, 0.36, and 0.55, respectively. It can be seen that the stress transfer rate

increases as the number of encasement layers increases. This is because the overlapping part of the multi-layer-encased geogrid increases the vertical modulus of the GESC while increasing the contact area between the column and soil, which makes the lateral restraint and lateral friction resistance of the column by the soil around the column increase, which leads to the increase in column modulus under the combined influence of multiple factors.

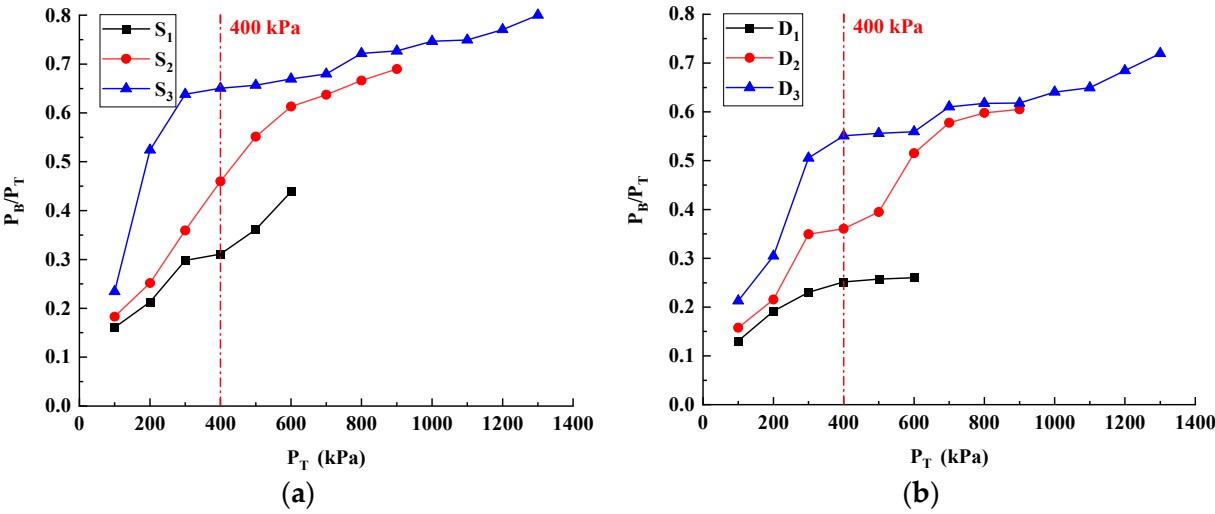

**Figure 18.** Real-time variation curve of GESC stress transfer rate under different numbers of encasement layers. (**a**) static loading. (**b**) dynamic loading.

## 4. Conclusions

In this study, GESC indoor lateral-limited compression tests were conducted to study the load-bearing performance and failure modes of a single GESC encased by a multi-layer geogrid under dynamic and static loading, and the following conclusions were mainly drawn:

1. The multi-layer encasement improved the vertical stiffness of the stone column, and the overlapping part of the external geogrid could fix the aggregate protruding due to the destruction of the internal geogrid, which improved the bearing capacity and delayed the failure of the stone column. Therefore, compared with stone column encased by a one-layer geogrid, a multi-layer geogrid-encased stone column has significantly improved bearing performance; in addition, a three-layer geogrid-encased stone column compared with a two-layer geogrid-encased stone column, under dynamic and static loading, had a vertical ultimate bearing capacity increased by 56.5% and 9.4%, respectively; the difference in the degree of improvement was obvious. Therefore, the encasement method should be reasonably selected according to the load type for the soft foundation treatment project. When a GESC is applied to the soft foundation treatment under static loading, two layers of encasement is the optimal number of layers for the stone column based on engineering feasibility and economy, while three layers of encasement are the optimal number of layers for the stone column when dynamic loading is considered;

2. Under dynamic and static loading, the stone columns all undergo the compression-density stage—linear elastic stage—yielding stage—failure stage; the GESC failure mode is bulging failure, and the main radial strains appear at the same height, so the effect of the difference between dynamic and static loading can be ignored. However, compared with static loading, the lateral restraint effect on the stone column is weaker under dynamic loading, and the radial strain is more uniform and smaller along the column height;

3. Compared with static loading, the vibration effect of dynamic loading disturbs the soil around the column and weakens the lateral restraint effect on the column, which indirectly leads to a decrease in the modulus of the GESC and a decreased stress

transfer rate. Meanwhile, with the increase in encasement layers, the overlapping of multi-layer geogrid increased the vertical modulus and lateral restraint, which increased the stress transfer rate and reduced the difference between dynamic and static loading on column failure;

4. Under the same conditions, the dynamic loading was relative to the static loading, and the vertical ultimate bearing capacity of the column encased by one-layer and two-layer geogrid under the dynamic loading was 5.0~20.3% lower than the static loading. When designing and serving soft foundation treatment projects such as expressways and high-speed railways, it is necessary to consider the impact of traffic loads on the bearing capacity of the stone column, and appropriately increase the design value of the bearing capacity of the stone column.

**Author Contributions:** Methodology, J.W.; Investigation, Y.Z.; Resources, S.H.; Data curation, B.K.; Writing—original draft, B.K.; Supervision, J.W. All authors have read and agreed to the published version of the manuscript.

**Funding:** This research was funded by the National Natural Science Foundation of China (No. 41962017), The Basic Ability Enhancement Program for Young and Middle-aged Teachers of Guangxi (No. 2023KY0731), the Natural Science Foundation in Guangxi Province of China (No. 2022GXNSFDA035081), the Natural Science Foundation of Guangxi (No. 2020GXNSFBA297163), the High-Level Innovation Team and Outstanding Scholars Program of Guangxi Institutions of Higher Learning of China (GuiJiaoRen-Cai[2020]6), the Innovation Project of Guangxi Graduate Education (No. YCSW2021310), and the Scientific Research Project of Hezhou University (2021ZZZK13).

**Institutional Review Board Statement:** Not applicable.

**Informed Consent Statement:** Not applicable.

**Data Availability Statement:** The data presented in this study are available on request from the corresponding author.

**Conflicts of Interest:** The authors declare no conflict of interest.

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
