# Peer review of "Study on Bearing Capacity and Failure Mode of Multi-Layer-Encased Geosynthetic-Encased Stone Column under Dynamic and Static Loading"

_sustainability, doi:10.3390/su15065205_

Round 1

Reviewer 1 Report

The manuscript focuses on the bearing capacity difference of multi-layer encased geosynthetic-encased stone column under dynamic and static loading. By analyzing the load-bearing performance and failure modes of single geosynthetic-encased stone column, the bearing mechanism and failure mode of the stone column was investigated. This topic is of interest to the journal. The findings are compelling and beneficial for the practitioners in this field. I found the paper was well-structured and informative. I gladly recommend it for publications in the journal. However, there are still some minor concerns:

1.      In the introduction: What is the innovation of this article? In addition, the author has not comprehensively reviewed the literature in this research field. The following documents are recommended:

Particle breakage behaviors of a foundation filling material on island-reefs in the South China Sea under impact loading. Bulletin of Engineering Geology and the Environment. 2022, 81 (9): 345. DOI: 10.1007/s10064-022-02844-3.

Particle breakage mechanism and particle shape evolution of calcareous sand under impact loading. Bulletin of Engineering Geology and the Environment. 2022, 81 (9): 372. DOI: 10.1007/s10064-022-02868-9.

2.      The English of the paper should be improved. There are some places where grammar should be corrected. Please carefully check and correct throughout the manuscript.

3.      Page 2, Lines 23-25: The abstract lacks the quantitative presentation of main results. The authors are required to add more specific information.

4.      Page 13, Line 244-245: Please check whether the loading label in Fig.6 (b) is wrong, and if so, please modify it.

5.      Page 16, Line 313: Please indicate the type of load applied in Figure 9 to avoid unnecessary misunderstandings.

6.      Page 27, Lines 501-502: Please mark the two pictures separately. Make it easier for readers to understand.

7.      Page 28, Line 534: The “the column encased by 1 and 2-layer geogrid” is not clear enough. Please modify the expression to make it more clear and easy to understand.

Author Response

Response to Reviewer 1 Comments

Dear reviewers:

 Thank you very much for your letter of regarding our manuscript entitled Study on bearing capacity and failure mode of multi-layer encased geosynthetic-encased stone column under dynamic and static loading (sustainability-2210843). Your review comments are very important and valuable to us. We have carefully read the review comments. Based on your review comments, we have improved the manuscript to meet your requirements for the manuscript to the greatest extent. The following are the changes in the manuscript.

The black part is your review opinion, and the blue part is our revision part.

Point 1: In the introduction: What is the innovation of this article? In addition, the author has not comprehensively reviewed the literature in this research field. The following documents are recommended: 

Particle breakage behaviors of a foundation filling material on island-reefs in the South China Sea under impact loading. Bulletin of Engineering Geology and the Environment. 2022, 81 (9): 345. DOI: 10.1007/s10064-022-02844-3.

Particle breakage mechanism and particle shape evolution of calcareous sand under impact loading. Bulletin of Engineering Geology and the Environment. 2022, 81 (9): 372. DOI: 10.1007/s10064-022-02868-9.

Response 1: Thanks for the valuable suggestions made by the reviewer, the Article recommended has been cited. 

The innovation of this article: this study carried out multiple sets of GESC large-scale indoor dynamic and static loading comparative model tests encased by multi-layer geogrids, the bearing mechanism of multi-layer encased geosynthetic-encased stone column was further understood by comparing the bearing difference of stone column under dynamic and static loading, which provided the experimental basis for GESC engineering design.

Point 2: The English of the paper should be improved. There are some places where grammar should be corrected. Please carefully check and correct throughout the manuscript.

Response 2: Thanks for the valuable suggestions made by the reviewer, the manuscript has been carefully checked and corrected.

Point 3: Page 2, Lines 23-25: The abstract lacks the quantitative presentation of main results. The authors are required to add more specific information.

Response 3: Thanks for the valuable suggestions made by the reviewer, the quantitative presentation has been added.

Point 4: Page 13, Line 244-245: Please check whether the loading label in Fig.6 (b) is wrong, and if so, please modify it.

Response 4: Thanks for the valuable suggestions made by the reviewer, the Error has been corrected.

Point 5: Page 16, Line 313: Please indicate the type of load applied in Figure 9 to avoid unnecessary misunderstandings.

Response 5: Thanks for the valuable suggestions made by the reviewer, the type of load applied has been added

Point 6: Page 27, Lines 501-502: Please mark the two pictures separately. Make it easier for readers to understand.

Response 6: Thanks for the valuable suggestions made by the reviewer, two pictures have been added notes.

Point 7: Page 28, Line 534: The “the column encased by 1 and 2-layer geogrid” is not clear enough. Please modify the expression to make it more clear and easy to understand.

Response 7: Thanks for the valuable suggestions made by the reviewer, the expression has been modified.

Reviewer 2 Report

The evaluated paper presents an attempt to explain the mechanism of the method of joining geosynthetics "overlap" in the case of making a stone column, which was tested under dynamic and static loads as part of the tests. The results of the conducted research are promising and have practical significance for application in geoengineering. The evaluated paper contains a sufficient introduction to the analyzed issues and references to the effects of previous, similar studies conducted by other researchers. It contains a detailed description of the research methodology used, the course of the research, and the analysis of the results obtained. In my opinion, the most valuable part of the work was the research carried out by the authors of the work on large-size models, generating conditions most similar to real ones occurring during the implementation of engineering goals, and a wide range of tests performed. The presented conclusions are specific, detailed and concern both favourable and less desirable effects.

 Comments:

• In Table 1, there is no need to repeat the same column dimensions, particle size or loading method;

• Table 2 should consistently say "Water content";

• Figure 2 is very dark, so it's hard to read, I recommend correcting it in an appropriate graphics program;

• Arrangement of drawings should be consistent, i.e. next to each other. Note applies to figures 3 and 4, 9a and 9b, 12.

Author Response

Response to Reviewer 2 Comments

Dear reviewers:

Thank you very much for your letter of regarding our manuscript entitled Study on bearing capacity and failure mode of multi-layer encased geosynthetic-encased stone column under dynamic and static loading (sustainability-2210843). Your review comments are very important and valuable to us. We have carefully read the review comments. Based on your review comments, we have improved the manuscript to meet your requirements for the manuscript to the greatest extent. The following are the changes in the manuscript.

The black part is your review opinion, and the blue part is our revision part.

Point 1: In Table 1, there is no need to repeat the same column dimensions, particle size or loading method.

 Response 1: Thanks for the valuable suggestions made by the reviewer, Table 1 has been modified as recommended.

Point 2: Table 2 should consistently say "Water content".

Response 2: Thanks for the valuable suggestions made by the reviewer, "water content" has been revised as "Water content".

Point 3: Figure 2 is very dark, so it's hard to read, I recommend correcting it in an appropriate graphics program.

Response 3: Thanks for the valuable suggestions made by the reviewer, it has been corrected according the suggestion to make it easy to read.

Point 4: Arrangement of drawings should be consistent, i.e. next to each other. Note applies to figures 3 and 4, 9a and 9b, 12.

Response 4: Thanks for the valuable suggestions made by the reviewer, It has been corrected according the suggestion.
